# Negative Impacts of Sleep–Wake Rhythm Disturbances on Attention in Young Adults

**DOI:** 10.3390/brainsci12121643

**Published:** 2022-11-30

**Authors:** Zijun Li, Shimin Fu, Fan Jiang, Weizhi Nan

**Affiliations:** 1Department of Psychology and Center for Brain and Cognitive Sciences, School of Education, Guangzhou University, 230 Wai Huan Xi Road, Guangzhou Higher Education Mega Center, Guangzhou 510006, China; 2Interdisciplinary Center Psychopathology and Emotion Regulation (ICPE), Department of Psychiatry, University Medical Center Groningen, University of Groningen, 9700 RB Groningen, The Netherlands; 3Department of Sleep Disorders, The Affiliated Brain Hospital of Guangzhou Medical University, Guangzhou 510180, China

**Keywords:** sleep–wake rhythm disturbances, sleep regularity, attention network test, attention, actigraphy

## Abstract

Sleep–wake rhythm disturbances have a negative impact on attention. However, how it affects attention and whether the restoration of regular rhythms can restore attention are unclear. This study aims to explore the effects of sleep–wake rhythm disturbances on three subfunctions of attention (alertness, orientation, and executive control) and the restoration of regular rhythms on these functions. Twenty-one participants in the experimental group (who experienced sleep–wake rhythm disturbances for at least one month; aged 18–26) were required to sleep regularly following a sleep schedule, whereas 20 participants in the control group (who maintained regular sleep for at least three months; aged 19–22) received no manipulation of their sleep. All participants were assessed using the attention network test three times in six days. All of them wore spectrum activity monitors and kept sleep diaries every day. The results showed that the effects of alertness and executive control in the experimental group were significantly lower than those in the control group. After five days of regular sleep, the difference in the alertness effect between the two groups significantly decreased. These results suggested that under natural conditions, sleep–wake rhythm disturbances could negatively influence alertness and executive control, and a short period of restoring a regular rhythm has a recovery effect on alertness.

## 1. Introduction

The impact of sleep disturbance on humans is far more than just feeling tired during the day. The quality of sleep affects all aspects of life, such as human memory consolidation, attention, cardiovascular and cerebrovascular diseases, the immune system, cancer cells, and sexual function [1]. Up to 76% of youths reported that they experienced sleep problems [2]. Modern lighting, multiple-night activities, and the use of electronic devices delay the sleep–wake rhythm and shorten sleep time [3,4]. These factors interrupted youths’ sleep–wake rhythm and reduced sleep quality.

Sleep–wake rhythm disturbances, also known as circadian rhythm sleep disturbances, indicate a lack of a defined sleep pattern. The intrinsic body clock regulates our daily biological and physical activities, such as body core temperature and sleep-wake behaviors [5,6]. In recent years, numerous studies have associated students’ academic performance with sleep quality. Attention, as a basic cognitive function, is one key issue [7,8,9,10,11,12,13,14]. Most studies have shown that sleep deprivation impairs attention [15]. Youths with chronic sleep deprivation may have impaired daytime functions, attention deficits, or unsatisfactory learning performance [16]. A long-term lack of sleep would affect normal visual function, and participants required a longer time to find target stimuli and judge colors [17].

Moreover, attention could be further decomposed into three subfunctions: alertness, orientation, and executive control [18]. “Alertness” refers to a state in which an individual maintains a high degree of mental concentration to receive stimulation. “Orientation” refers to the ability to quickly find the required information from a large amount of input information. “Executive control” refers to the ability to suppress irrelevant stimuli and select target stimuli correctly when there is conflicting information. Previous studies have shown that subfunctions are influenced by sleep quality, especially alertness and executive control. Peters et al. found that after 5 h of sleep restriction, participants showed reduced alertness compared with those in the control group [19]. Jugovac and Cavallero found that executive control efficacy significantly decreased after one night of sleep deprivation [20].

In addition to sleep deprivation, sleep–wake rhythm disturbances are another of the most common sleep disorders among contemporary youths [21]. Sleepers with irregular rhythm have poorer academic performance and delayed circadian rhythm than sleepers with regular rhythm [10]. Delayed circadian rhythms lead to disturbances or even loss of sleep structure, which would further lead to disruptions in attention [22].

It is widely accepted that sleep deprivation will disrupt attention in youths, but whether irregular rhythms in natural conditions will also cause a decline in attention in youths still lacks evidence [8,23,24,25]. There is little evidence of the effect of restoring sleep–wake rhythm on the restoration of attention. To address these issues, in this study, we adopted the characteristics of a randomized controlled trial and selected college students with sleep–wake rhythm disturbances as the experimental group and those with regular rhythms as the control group; repeated measures analyses were used to study the effects of sleep–wake rhythm disturbances on the attention of youths and further explore the effects of sleep–wake rhythm restoration on attention. According to the previous studies mentioned above, we hypothesized the following: (1) Similar to sleep deprivation, sleep–wake rhythm disturbances will also negatively affect attention, especially in terms of alertness and executive control. (2) Restoring sleep–wake rhythm can restore different dimensions of attention.

## 2. Materials and Methods

### 2.1. Participants

Participants were recruited through online advertisements from Guangzhou Higher Education Mega Center: a large educational area containing eleven universities, and all of them participated voluntarily. The inclusion criteria were as follows: 1. Participants could guarantee that they could ensure a stable sleep schedule during the experimental period. 2. Participants did not have a day-night shifted work schedule or travel across time zones within three months. 3. Participants had no confirmed organic disease. 4. Participants were not receiving sleep therapy or medication. 5. Participants did not have moderate or above sleep hypopnea apnea syndrome. 6. Participants did not have restless legs syndrome. 7. Participants had not taken psychotropic drugs within three months. 8. Participants had not taken corticosteroids within three months. 9. Participants had no diagnosed sleep disorders (Pittsburgh Sleep Quality Index, PSQI scores less than 13 points, the cutoff point of diagnosed patient in PSQI-Chinese version is >12.) [26]. 10. Participants had no moderate to severe depression (Beck’s Depression Inventory (BDI) scores less than 14). 11. Participants had no severe anxiety (Self-rating Anxiety Scale (SAS) scores less than 50). 12. Participants had no history of mania or other psychotic disorders. 13. Participants had no alcohol use disorder. 

The participants were not women who were in their menstrual period. 15. The participants were not women who were pregnant or breastfeeding.

The PSQI questionnaire was completed online, and a total of 329 valid questionnaires were collected. In the first round of screening, the presence of sleep–wake rhythm disturbances were based on the Sack’s description of clinical symptoms [27] and combined with the living habits of college students. Participants who adjusted for differences in bedtime every day within half an hour and usually went to bed before 24:00 for at least three months were included as candidates for the control group. In total, 20 healthy adult participants (female, 11) aged 19–23 years old were selected. Participants whose daily bedtime difference was greater than 2.5 h and who usually went to bed after 1:30 am for at least one month were included as candidates for the experimental group. Twenty-four participants (females, 10) aged 18–26 years old were selected. In the second round of screening, participants with PSQI scores equal to or greater than 13 were excluded. In the third round of screening, participants attended one-on-one structured interviews. In the interview, the interviewer asked and confirmed the prior sleep schedule of participants and allocated them into a control group or an experimental group based on the Sack’s description of clinical symptoms about sleep–wake rhythm disorder [27]. In the interview, participants completed a BDI and SAS, and the scores could not be greater than 14 and 50, respectively. All participants who did not meet any of the inclusion criteria were excluded.

Each participant voluntarily enrolled and signed an informed consent form prior to the experiments. This study was approved by the institutional review board of the Educational School, Guangzhou University with the protocol number GZHU2020017.

### 2.2. Study Design

According to their prior sleep schedule and clinical symptoms, participants were divided into the experimental group and the control group based on the Sack’s description of clinical symptoms [27]. All participants completed the Attention Network Test (ANT) three times (pretest, midtest and posttest) [28,29]. The participants in the experimental group received sleep restriction based on cognitive behavioral therapy for insomnia (CBT-i) to restore a regular sleep schedule after the pretest. The control group received no intervention regarding sleep schedules but followed their past routines. The participants were required to complete a sleep diary every day and wore an Actiwatch. The study was a repeated measures design of 2 (group: experimental group, control group) × 3 (time point: pretest, midtest, and posttest). The intragroup variable was time point, and the intergroup variable was the characteristics of sleep regularity in that group. The outcome variables were the effect of alertness, orientation and executive control in ANT.

### 2.3. Apparatus, Materials, and Procedure

#### 2.3.1. Measurement of Sleep Parameters

We used an Actiwatch(MAMBO2, Lifesense Medical Electronics Co., Ltd., Zhongshan, China) to collect participants’ biorhythms in real time, automatically calculate sleep efficiency and sleep duration, and record physiological characteristics, such as the number and duration of night awakening and morning awakening heart rate. Actigraphy is a reliable and widely used tool for collecting data on sleep parameters and circadian rhythm [27,30]. These data were uploaded to a mobile application for viewing and data cleaning. In addition, we also used a sleep diary to enable participants to independently record their sleep and daytime activities. The content of the sleep diary was similar to that of the Actiwatch. However, factors such as the amount of daily exercise, caffeine intake, alcohol intake, emotion and the use of electronic devices before going to bed were also included. The researcher could review a participant’s sleep diary to regulate and inspect their sleep schedule. In situations where the data from the actigraphy did not match their sleep diary, the participants were required to report and explain the difference.

#### 2.3.2. Measurement of Attention

E-Prime 2.0 (for Windows, Psychological Software Tools, Inc., Pittsburgh, PA, USA) was used to program the ANT, which is a combination of the cued response time (RT) task and the flanker task [28,29]. This task minimized the influences of working memory, semantic processing, emotional perception, and other cognitive functions due to its simple design [31].

The participants were seated in a sound-attenuated chamber approximately 60 cm away from a cathode ray tube (CRT) monitor (resolution: 1024 × 768 pixels, vertical refresh rate: 85 Hz), and their eyes were the same height as the center of the monitor. A 12-trial full-feedback practice block was provided, followed by two experimental trial blocks. Each experimental block consisted of 96 trials that were displayed in random order. A “take a rest” signal was shown on the screen between blocks.

Note that the three functions of alertness, orientation, and execution control are all operationally defined by comparing the RT under different conditions, and the difference between two corresponding control conditions was determined by a certain attention function [28]. In the ANT (see Figure 1), there were three types of cues: no cues, central cues, and spatial cues. The spatial cues are divided into two types: invalid cues (where the cues appear in different sites from the target stimulus) and valid cues (where the cues appear at the same site as the target stimulus). There are two types of stimuli: congruent stimuli and incongruent stimuli. Based on these six different conditions, 8 different types of experimental trials can be produced.

The following is the expression for calculating the attention functions [28]:Alertness = RT _(no cue)_ − RT _(central cue)_(1)
Orientation = RT _(central cue)_ − RT _(spatial cue)_(2)
Executive Control = RT _(incongruent stimulus)_ − RT _(congruent stimulus)_(3)

The above formulas show that the RT difference between the no cue and central cue conditions is generated by alertness. The RT difference between the central cue and the spatial cue conditions is generated by orientation. The RT difference between incongruent stimuli and congruent stimuli is generated by executive control. Participants were required to respond to the direction indicated by the middle arrow.

Fan and his colleagues tested the reliability and validity of the ANT with 40 adults [29]. The research results showed that the alertness test-retest reliability coefficient was 0.52, the orientation test-retest reliability coefficient was 0.61, the executive control test-retest reliability coefficient was 0.77, and the repeat test reliability coefficient was 0.87. In the correlation analysis among the three functions, no obvious correlation was noted among alertness, orientation, and executive control.

#### 2.3.3. Procedures

Research found that 6 consecutive nights of sleep extension can benefit sleepy participants in daytime alertness [32]. To follow the strict school schedule (from Monday to Friday) of our participants, we adopted a 5-night training period, which is logically more practical. In this way, dropouts due to college weekend activities were reduced. A baseline was measured on the first day, and the participants came to the laboratory in the morning (approximately 2–3 h after getting up) for the first test of the ANT. The interviewer distributed an Actiwatch and individually negotiated with each participant to formulate the sleep schedule that needed to be strictly executed in the following 5 days. For participants in the experimental group, the sleep schedule needed to meet three conditions: 1. It should not be day–night shifted from the previous schedule. That is, if the participant is an extremely night-type person, then the sleep schedule should be maintained in night type. 2. Try to ensure that the sleep duration is approximately 7 h. 3. Ensure sleep regularity. That is, based on the Sack’s description of clinical symptoms, participants in the experimental group must schedule their bed-time within half an hour throughout the whole experimental period [27,33]. These participants had irregular bed times that varied from 2.5 h prior to the intervention. Based on the CBTi protocol, we used stimulus control to rebuild the connection between bed and sleep. For example, participants were asked to prevent any behaviors that are not related to sleep when on bed. Therefore, the conditioning between sleep and bed was strengthened [34]. Sleep restriction was also used to rebuild the circadian rhythm of participants. Participants with irregular sleep schedules were trained to maintain a regular sleep schedule with a concrete and stable bed time and wake time. Napping longer than 1 h before 18:00 and napping after 18:00 was prevented.

For participants in the control group, the only requirement for the schedule was to keep the same pattern from the previous schedule. All participants signed an informed consent form and commitment, in which they promised to follow the experimental regulations during the experiment.

On the second and third days, participants followed the sleep schedule, completed sleep diaries every day, and wore the Actiwatch. The sleep diary recorded the time of using electronic products before going to bed by minutes, daytime alcohol intake, daytime exercise, daytime caffeine intake, sleep duration, sleep efficiency, work efficiency, and emotion during that day. On the fourth day of the experiment, participants came to the laboratory at the same time as the first test (2 ± 1 h after getting up) to take the ANT mid-test. Then, they continued to follow the procedure. On the sixth day, the participants came to the laboratory again at the same time to take the ANT posttest (see Figure 2).

#### 2.3.4. Statistical Analysis

R-Studio 1.1.463 (for Mac, Posit PBC, Boston, MA, USA) and SPSS Statistics 21.0 (for Mac, IBM, Armonk, New York, USA) were used for data analysis. Three repeated measures analyses of variance (ANOVAs) of a 2 (control group and experimental group) × 3 (pretest, midtest, posttest) design were conducted for the three subfunctions of attention. Three repeated-measure MANOVA of a 2 (control group and experimental group) × 3 (pretest, midtest, postest) × 2 (corresponding cues) were conducted for the original RT for further analysis. The assumption of Mauchly’s test of sphericity for functions orientation and executive control were violated; therefore, their values were corrected by the Greenhouse-Geisser correction.

## 3. Results

### 3.1. Characteristics of Participants

The experimental group and the control group were approximately the same age (*t*_39_ = 0.9, *p* = 0.36). During the experiment, the sleep duration and sleep efficiency of the two groups met the experimental criteria (see Table 1). The sleep duration was greater than 7 h, and the sleep efficiency was greater than 85%, which indicates satisfactory sleep quality [35]. Neither sleep duration nor sleep efficiency showed a significant difference (*t*_203_ = −0.4, *p* = −0.7; *t*_203_ = −0.7, *p* = 0.5) between the groups. In the experimental group, the average BDI score was 6.8 ± 5.9, and the average SAS score was 42.0 ± 4.8, which were both within the normal range. The average scores of the control group on the two scales were slightly lower than those of the experimental group at 4.5 ± 5.4 (*t*_39_ = 1.3, *p* = 0.2) and 35.9 ± 4.2 (*t*_39_ = 3.6, *p* = 0.001), respectively. The participants who showed chronic irregular rhythms had higher scores on anxiety but were still under the threshold of clinical criteria. For convenient reading, the degree of freedom is reported as equal variances assumed while their values are in fact corrected by unequal variances.

The control group consisted of 20 healthy adult participants (female, 11) aged 19–23 years (20.2 ± 1.1). None of the participants stayed up late, stayed up overnight, or travelled across time zones within the previous month. The average daily sleep duration was at least 7 h, and the daily time to fall asleep and wake up was stable (standard deviation (SD) < 0.5 h). During the experiment, all participants followed the past routines to sleep without any intervention and ensured that there was no significant change in sleep schedule within the five days. The average sleep duration in the control group was 7.6 h, and the sleep efficiency was 92%.

The experimental group included a total of 21 participants (females, 10) aged 18–26 years (20.6 ± 1.7) with chronic (at least one month) sleep–wake rhythm disturbances. The World Health Organization defines youths as individuals aged between 15 and 24. Only one participant (aged 26) was older than 24-years-old because she started her study program two years later than her counterparts. Since our enrollment criteria is strict and her daily routine is the same with others, we did not exclude the valuable data of this participant from the experimental group. All participants did not have a particular time at which they fell asleep at night and woke up in the morning, and there was a variance ranging from 1.5 to 4 h. During the experiment, the participants followed the sleep schedule. The average sleep duration was 7.5 h, and the average sleep efficiency was 91%. The exact sleep time and waketime of each participant are provided in Appendix A.

### 3.2. Effects of Sleep–Wake Rhythm Disturbances on Attention

For the alertness function (see Figure 3 and Table 2), the main effect of time point was not significant (*F* (2, 78) = 1.441, *p* = 0.243, η^2^ = 0.036), indicating that there was no significant difference between the results at the three time points and that the influence of learning effect could be excluded in the alertness task. The main effect of group was significant (*F* (1, 39) = 67.046, *p* < 0.001, η^2^ = 0.632), indicating that the effect of alertness in the control group (22 ms) was significantly greater than that in the experimental group (−14 ms). Regarding the original RT, the interaction between the cue (center cue and no cue) and group was significant (*F* (1, 67) = 1.719, *p* < 0.001), indicating that the experimental group was slower on the task under the center cue and faster on the task under no cue when compared with the control group. The interaction between time points and groups was significant (*F* (2, 78) = 3.564, *p* = 0.033, η^2^ = 0.084), indicating that the experimental group and the control group had significantly different trends at different time points. The simple effect analysis showed that alertness in the midtest and posttest of the experimental group was significantly improved (pretest: −19 ms, midtest: −19 ms, posttest: −6 ms, *p* = 0.037), whereas participants in the control group showed no significant difference throughout the whole experiment (pretest: 18 ms, midtest: 28 ms, posttest: 19 ms). For the original RTs, the interaction between time*cue*group was significant (*F* (2, 38) = 4.772, *p* = 0.014), indicating that the difference in RTs between cues in the experimental group was larger than that in the control group due to a significantly faster RT on the task under the center cue (15 ms). These findings showed that the participants in the experimental group were affected by sleep training. After five days of regular sleep, alertness was significantly improved.

For the function of orientation (see Figure 3 and Table 2), the main effect of time points was significant (*F* (2, 78) = 4.789, *p* = 0.017, η^2^ = 0.109). A further analysis of original RTs also showed that all participants responded significantly faster during the three time points (*F* (2, 38) = 18.862, *p* < 0.001), indicating that there may have been learning effects on the orientation task. The main effect of group (*F* (1, 39) = 2.196, *p* = 0.146, η^2^ = 0.053) and the interaction between the three time points and groups (*F* (2, 78) = 2.008, *p* = 0.150, η^2^ = 0.049) were not significant, indicating that sleep–wake rhythm disturbances did not influence orientation.

For the executive control function, the main effect of time points was significant (*F* (2, 78) = 24.276, *p* < 0.001, η^2^ = 0.384). A further analysis of original RTs also showed that all participants responded significantly faster during the three time points (*F* (2, 38) = 16.771, *p* < 0.001), indicating that there may have been learning effects in the executive control task. The main effect of groups was significant (*F* (1, 39) = 4.224, *p* = 0.047, η^2^ = 0.098, see Figure 3). Simple effect analysis showed that the effect of executive control in the experimental group (75 ms) was significantly smaller than that in the control group (90 ms). Further analysis of the original RTs showed that the interaction between the cue (congruent cue and incongruent cue) and group was significant (*F* (1, 39) = 4.224, *p* = 0.047), indicating that the experimental group was generally slower on the task under the congruent cue condition and faster on the task under the incongruent cue condition than the control group. These findings indicated that sleep–wake rhythm disturbances affected the effect of executive control by prolonging the RTs of tasks under the congruent cue. However, the interaction between group and time points was not significant (*F* (2, 78) = 1.408, *p* = 0.251, η^2^ = 0.035, see Figure 3), indicating that a regular sleep rhythm for five days was not sufficient to significantly restore impaired executive control.

## 4. Discussion

In the present study, a comparison was made between healthy controls and people with sleep–wake rhythm disturbances in youths. Our results showed that disturbances in the sleep–wake rhythm result in substantial disruptions in alertness and executive control but have no influence on orientation. After five days of regular sleep training, the participants with sleep–wake rhythm disturbances could effectively recover alertness to some extent, but there was no significant recovery in executive control.

### 4.1. Effects of Sleep–Wake Rhythms on the Three Subfunctions of Attention

A large number of previous studies have found that workers who reverse day and night and people who frequently travel across time zones have significant decreases in alertness because the sleep–wake rhythm does not match the circadian rhythm formed by their homeostasis system [36]. Our research results are consistent with these previous studies. In addition, a very large difference was noted between groups. Participants who maintained regular schedules for a long time scored positive on alertness (21 ms), whereas participants with chronic sleep–wake rhythm disturbances scored negative on alertness (−14 ms). The original RTs for the center cue were significantly slower than those for the no cue in the experimental group. This finding indicated that the cue appearing before the target (center cue) failed to increase but distracted the attention of the participants with chronic sleep–wake rhythm disturbances. Moreover, after 5 days of sleep training, participants with chronic sleep-wake disturbances showed significant increases in alertness. Although the score remained negative, it gradually approached zero, and the difference from healthy controls was substantially decreased. Our finding is partially consistent with the study of Belenky (2003) They found no recovery effect after 3-night sleep recovery in the 5–7 h/night sleep group. In the 3 h/night sleep group, alertness recovered rapidly following the first night of recovery sleep, although the recovery was incomplete [37]. These results implied that if it takes longer to restore regular sleep–wake rhythms, the alertness of individuals with sleep–wake rhythm disturbances might return to the same levels as healthy controls.

In terms of orientation, this study did not find a significant difference between the two groups. This finding is consistent with a previous study that used the ANT-I for attention assessment. Waldon et al. found that poorer sleep had no significant impact on orientation in children with attention-deficit/hyperactivity disorder (ADHD) or typical development [38]. In contrast, some studies found no significant difference in the selective attention of participants undergoing sleep restriction by digit symbol substitution tasks, whereas another study using the same paradigm with participants undergoing sleep deprivation found significantly lower accuracy on selective attention [7]. This finding implied that different degrees of interruption to sleep–wake rhythm had different effects on attention. In our study, due to learning effects, the participants with regular sleep schedules showed significantly faster RTs during the three test time points, whereas participants with chronic sleep–wake rhythm disturbances did not show learning effects. It might be explained that a mild degree of chronic sleep–wake rhythm disturbances might not be enough to affect orientation, but it limited their ability to improve orientation through practice.

Regarding executive control, a significant difference was noted between the two groups that demonstrated that chronic sleep–wake rhythm disturbances impaired executive control. Participants who had regular sleep schedules had a significantly greater effect in the pretest, which indicated a better function on the executive control task compared with those with sleep–wake rhythm disturbances [29]. A previous study also found that sleep irregularity caused lower scores in executive control by applying the Trail-Making test [39]. In addition, disrupting participants’ 24-h rhythms has been shown to result in a significant decrease in participants’ executive function [40]. In our research, however, although the two groups of participants exhibited significant differences in the effect of executive control, trends in their changes over time followed a similar pattern. Therefore, this improvement might be caused by learning effects rather than a restoration of the sleep–wake rhythm. A previous study also found that learning effects in executive control are clearly apparent because the incongruent condition would be learnt in the process, especially in young adults [41]. Our findings provide consistent and relevant evidence for this research. For further exploration of the restoring effects, we need to extend the period of sleep training. Participants with sleep–wake rhythm disturbances can maintain a regular sleep–wake rhythm for a longer time so that a larger effect of executive control might be found.

### 4.2. Control of Irrelevant Variables

Over the course of the experiment, to maximize the reliability of the results and reduce the influence of other irrelevant variables, we arranged the time for all participants to perform the ANT to be within 1–3 h after waking up. The time for participants in the experimental group to complete the ANT was 9:30–11:30 am; the time for those in the control group to complete the ANT was 8:30–10:30 am. Tonetti used activity monitors to detect daytime activity rhythms of 32 healthy participants and reported a double peaked curve in human daytime activity levels at approximately 10 am and 8 pm under normal conditions [42]. Therefore, performing the ANT within 1–3 h after waking up not only ensured that the participants were performing under the best conditions but also avoided bias from lunch breaks or caffeine intake.

To ensure that the experimental control was factual and effective, we supplemented the use of a sleep diary to record sleep characteristics by using an activity monitor. The wrist-worn activity monitor can effectively record sleep–wake rhythms and various sleep characteristics, such as sleep duration, latency to fall asleep, time to fall asleep, and time to wake up. During the experiment, once we found that the sleep data presented by the wrist-worn monitor and the sleep diary were different, we immediately contacted the participants and asked for details regarding the specific situation to facilitate data removal and analysis.

### 4.3. Limitations and Future Directions

The implementation process was strictly controlled, and the study design has a certain degree of generalizability that can be widely applied in daily life. However, in particular situations, the results of this research still have limitations.

The main participants in this experiment were college students between 18 and 26 years old. In this state, human adaptability and resilience are relatively strong. The results of restoring alertness in a five-day period of regular sleep prescribed in this study may apply only to young individuals rather than older people.

Although this experiment used only the activity monitor to obtain the basic characteristics of sleep, the obtained results provided valuable ideas for subsequent research. We can extend the period of regular sleep–wake rhythm to explore whether alertness can be completely restored by restoring sleep regularity and whether executive control can show a recovery effect after a longer period of regular sleep–wake rhythm. Regarding the neurological mechanisms of the sleep–wake rhythm affecting attention, future studies are needed to monitor the sleep structure of participants. In particular, these studies could explore whether their sleep structure has changed, which sleep stage has a greater impact on attention, and what specific oscillation makes changes in attention.

## 5. Conclusions

This study generally validated the research hypothesis that in a population of youths, mild to moderate sleep–wake rhythm disturbances significantly weakened the alertness and executive control aspects of attention but had minimal effect on orientation. After restoring participants with chronic sleep–wake rhythm disturbances to a regular sleep schedule for five days, a significant restoration effect in alertness was noted, but no significant improvement in executive control was observed. Future studies could extend the period of sleep restoration to explore whether a complete recovery could be found.

## Figures and Tables

**Figure 1 brainsci-12-01643-f001:**
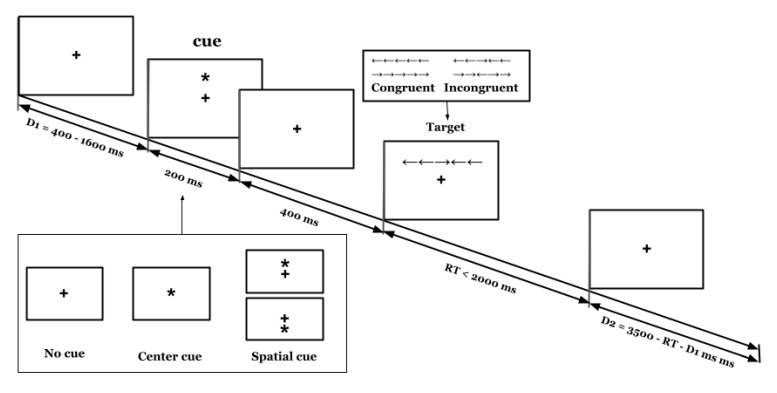
Schematic of the ANT. During the trial, a fixation cross was presented in the center of the screen and remained there the whole time. After a random duration D1 (400–1600 ms), a cue (none, center, or spatial cue) appears for 200 ms. After a 400-ms blank screen, a line of arrows with a target and flankers is presented (congruent or incongruent) above or under the fixation cross until the participant responds with a button press but for no longer than 2000 ms. Finally, a posttarget fixation period D2 is presented for a random duration.

**Figure 2 brainsci-12-01643-f002:**
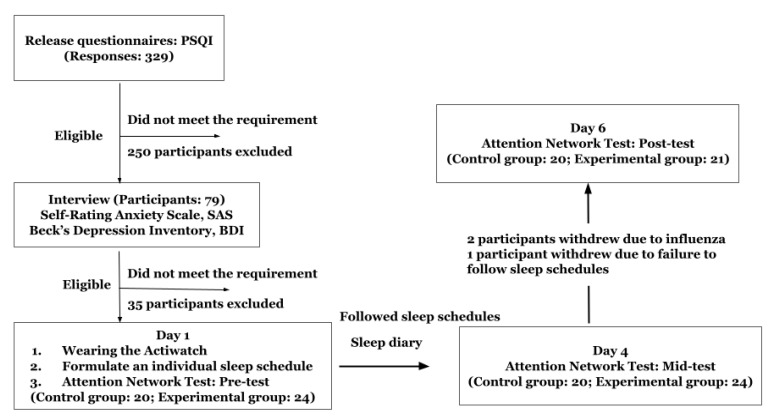
Schematic of the experimental procedure.

**Figure 3 brainsci-12-01643-f003:**
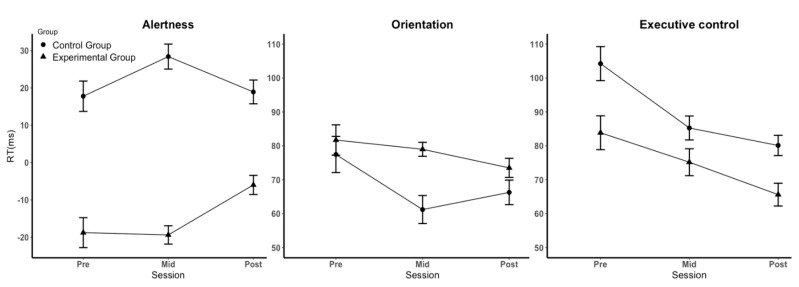
Effect (difference scores) with standard error (SE) for the three attentional subfunctions on RT.

**Table 1 brainsci-12-01643-t001:** Participant information.

	Experimental Group (*n* = 21)		Control Group (*n* = 20)		95% Confidence Interval	
	**Mean**	**SD**	**Mean**	**SD**	**Lower**	**Upper**
Age	20.6	1.7	20.2	1.1	−0.5	1.3
Sleep duration (h)	7.5	1.0	7.6	0.5	−0.3	0.2
Sleep efficiency	91%	0.1	92%	0.1	−0.03	0.02
BDI	6.8	5.9	4.5	5.4	−1.3	6.0
SAS	42.0	4.8	35.9	4.2	2.7	9.6

**Table 2 brainsci-12-01643-t002:** Response time in the attention task (RT in ms).

Condition	Group ^d^	Pretest (M ± SD)	Midtest (M ± SD)	Posttest (M ± SD)
No cue	1	638 ± 150	558 ± 36	553 ± 53
	2	654 ± 60	600 ± 50	571 ± 42
Central cue	1	657 ± 133	577 ± 38	558 ± 48
	2	636 ± 63	571 ± 42	552 ± 45
Valid cue	1	575 ± 143	498 ± 37	485 ± 54
	2	559 ± 57	510 ± 43	486 ± 42
Incongruent stimulus	1	665 ± 140	581 ± 42	564 ± 56
	2	666 ± 64	600 ± 47	574 ± 44
Congruent stimulus	1	581 ± 143	505 ± 34	499 ± 48
	2	561 ± 53	514 ± 38	494 ± 39
Alertness ^a^	1	−19 ± 26	−19 ± 16	−6 ± 16
	2	18 ± 26	28 ± 21	19 ± 20
Orientation ^b^	1	82 ± 29	79 ± 13	74 ± 18
	2	77 ± 34	61 ± 26	66 ± 23
Executive control ^c^	1	84 ± 32	75 ± 25	66 ± 21
	2	104 ± 32	85 ± 23	80 ± 19

^a^. Alertness = RT (no cue) − RT (central cue), ^b^. Orientation = RT (central cue) − RT (valid cue), ^c^. Executive control = RT (incongruent stimulus) − RT (congruent stimulus), ^d^. 1 = experimental group; 2 = control group.

## Data Availability

The anonymized data, stimuli, and preprocessing/analysis details are available upon request.

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
