# Peer review of "Negative Impacts of Sleep–Wake Rhythm Disturbances on Attention in Young Adults"

_brainsci, 2022, doi:10.3390/brainsci12121643_

Round 1

Reviewer 1 Report

Manuscript ID: brainsci-1982632

Title: "Negative impacts of sleep-wake rhythm disturbances on attention in young adults"

The aim of this study was to evaluate the possible negative effects of sleep-wake cycle disorders on attentional performances. To this end, 44 participants were enrolled online: 24 in the experimental group (42% females); 20 in the control group (55% females). To assess sleep-wake cycle, actigraphy and sleep diary were used. To assess attention, Attention Network Test was performed in three sessions: at baseline (pre-test), after four days (mid-test), and after other two days (post-test). Results are not so clear.

The topic of the present study is potentially interesting; however, the presentation is too awkward. It seems to me that Authors did not even re-read the manuscript before sending it. For example, many citations are missing in the bibliography (Eduard et al., 2017; Bugueno et al., 2017; Blunden et al., 2011; Kaufmann et al., 2013; Posner and Petersen, 1990; Sack et al., 2007; Lorenzo, 2018 and so on).

The introduction lacks a clear definition of what is meant by sleep-wake rhythm disturbances. This is a serious shortcoming considering that this is the topic of the work.

Usually, when an acronym is mentioned for the first time it should be expanded. This is the case of BDI and SAS at page 2.

In the participants section, no information is provided on the two selected groups.

In the study design section, Authors wrote about sleep restriction based on cognitive behavioural therapy for insomnia. However, it does not seem to me that a restriction of sleep was then practiced.

The actigraph is not used to record biorhythm; movement is a behaviour, not a biological variable.

The presentation of ANT seems a little bit confused.

The timing of the procedure is very mysterious. Why did Authors decide to assess again participants after only 4 days? And why after other two days? Which is the rationale?

Table 1 is not enough informative. The landmarks of sleep-wake cycle, i.e., bedtime and wake-up time, are totally missing. Moreover, control group showed a sleep efficiency lower in comparison with experimental group. This data is unexpected.

Looking at Table 2, a learning effect is present in all groups and in all tests. Why did not the authors comment on these data? Indeed, at page 8 the authors argue that the participants of the experimental group do not show the learning effect.

The conclusions are in fact missing.

Reviewer 2 Report

Manuscript ID: brainsci-1982632

Title: Negative Impacts of Sleep-Wake Rhythm Disturbances on Attention in Young Adults

Journal: Brain Sciences  

Abstract

1.     Authors should add the number of participants of the experimental and control group.

Introduction

1.     Page 1. Authors wrote “Peters et al. (2009) found that after 5 hours of sleep restriction, participants showed reduced alertness than those in the control group”. Authors should read and quote the following study: https://pubmed.ncbi.nlm.nih.gov/22044791/

Materials and Methods

1.     Footnote number 2 at page 1. Authors wrote “Here we used 13 as the cut off point of sever sleep disturbances.”. I am wondering whether this cut-off value was previously proposed in literature.

2.     Page 2. Authors wrote “This study was approved by the institutional review board of the Educational School, Guangzhou University.”. They should add the ethical committee report number and specify whether the informed consent was obtained.

3.     Page 3, measurement of attention sub-paragraph. Authors wrote “there were three types of cues: no cues, central cues, and spatial cues.”. It seems to me that the double cue condition is missing. Please also refer to the study by Fan and colleagues (2002): https://pubmed.ncbi.nlm.nih.gov/11970796/

4.     Page 4, measurement of attention sub-paragraph. Also according to Fan and colleagues (2002), the efficiency of the alerting network should be computed as the difference between the RT in no cue and RT in double cue condition.

5.     Page 4, measurement of attention sub-paragraph. When Authors discuss the learning effect at the ANT, they could also quote the following study: https://www.sciencedirect.com/science/article/pii/S0165027010002220

6.     Page 4, procedure subparagraph. Authors should replace “subjects” with “participants” or synonyms.

7.     Page 5, Figure 2. Authors should add the number of excluded participants at each step.

Results

1.     Page 5, characteristics of participants subparagraph. Authors wrote “The sleep duration was greater than 7 hours, and the sleep efficiency was greater than 85% which indicates a satisfied sleep quality”. Are there any available cut-off values for the activity monitor used in the current study?

2.     Page 5, Table 1. Authors should add the complete statistics with reference to each comparison between groups.

3.     Page 6, effects of sleep-wake rhythm disturbances on attention subparagraph. Separately for each attention network, Authors could perform a further statistical analysis on reaction times with group (two levels, experimental and control) as a between-subjects factor, time points (three levels, pretest, middle and post) and type of cue (two levels, e.g., no cue and double cue for the alerting network) as within-subjects factors.

4.     Page 6, effects of sleep-wake rhythm disturbances on attention subparagraph. Authors wrote “see Figure 3 A” but only Figure 3 is reported at page 7, without the reference to the letter (in this case, A).

5.     Page 7, legend of Figure 3. Authors wrote “Effect size (with standard error [SE]) for the three attentional subfunctions on RT.”. It seems to me that the effect sizes are not reported, only means and standard errors.

Round 2

Reviewer 1 Report

The revised version of the manuscript has improved, even though the Authors have not responded to all requests. For example, the conclusions remain very concise.  The definition of sleep restriction is truly original. The sleep data presented in the appendix are unusable as they refer to the individual nights of the subjects, it seems to me that a summary table of these data is still missing. 

Reviewer 2 Report

Manuscript ID: brainsci-1982632-R1

Title: Negative Impacts of Sleep-Wake Rhythm Disturbances on Attention in Young Adults

Journal: Brain Sciences  

General comment

Authors have adequately addressed my previous concerns. I do not have any further suggestions.

Specific comment

Some sentences, at pages 5, 8 and 9, are not readable.